# Leveraging Locality to Boost Sample Efficiency in Robotic Manipulation

**Tong Zhang**[1,2,3]    **Yingdong Hu**[1,2,3]    **Jiacheng You**[1]    **Yang Gao**[1,2,3*]

[1]Tsinghua University    [2]Shanghai Qi Zhi Institute    [3]Shanghai Artificial Intelligence Laboratory

{zhangton20,huyd21,yjc23}@mails.tsinghua.edu.cn,  gaoyangiiis@mail.tsinghua.edu.cn

**Abstract:** Given the high cost of collecting robotic data in the real world, sample efficiency is a consistently compelling pursuit in robotics. In this paper, we introduce SGRv2, an imitation learning framework that enhances sample efficiency through improved visual and action representations. Central to the design of SGRv2 is the incorporation of a critical inductive bias—*action locality*, which posits that robot's actions are predominantly influenced by the target object and its interactions with the local environment. Extensive experiments in both simulated and real-world settings demonstrate that action locality is essential for boosting sample efficiency. SGRv2 excels in RLBench tasks with keyframe control using merely 5 demonstrations and surpasses the RVT baseline in 23 of 26 tasks. Furthermore, when evaluated on ManiSkill2 and MimicGen using dense control, SGRv2's success rate is 2.54 times that of SGR. In real-world environments, with only eight demonstrations, SGRv2 can perform a variety of tasks at a markedly higher success rate compared to baseline models. Project website: `sgrv2-robot.github.io`.

**Keywords:** Robotic Manipulation, Sample Efficiency

## 1 Introduction

The creation of a versatile, general-purpose robot has long captivated the robotics community. Recent advances in imitation learning (IL) [1, 2, 3] have enabled robots to exhibit increasingly complex manipulation skills in unstructured environments. However, prevailing imitation learning techniques frequently require an abundance of high-quality demonstrations, the acquisition of which incurs substantial costs. This contrasts markedly with disciplines such as computer vision (CV) and natural language processing (NLP), wherein vast repositories of internet data are readily available for utilization. In this paper, we investigate methods to boost **sample efficiency** in robotic manipulation by developing improved visual and action representations.

In machine learning, introducing inductive bias is a standard strategy to enhance sample efficiency. For instance, CNNs [4, 5] inherently embed spatial hierarchies and translation equivariance in each layer, while RNNs [6] and LSTMs [7] incorporate temporal dependencies in their architecture. In the realm of robotic manipulation, a critical inductive bias is **action locality**, which posits that a robot's actions are predominantly determined by the target object and its relationship with the surrounding local environment. However, previous studies on representation learning for robotic manipulation have not effectively leveraged this bias. Typically, these studies [8, 9, 10, 11] aim to develop a global representation that encapsulates the entire scene, which is then directly employed to predict robot actions. These approaches have demonstrated notably low sample efficiency. As depicted in Figure 1, reducing the number of demonstrations from 100 to 5 leads to a substantial decrease in the performance of previous works such as SGR [10], which seeks to capture both semantic and geometric information in a global vector.

---

*Corresponding author

8th Conference on Robot Learning (CoRL 2024), Munich, Germany.

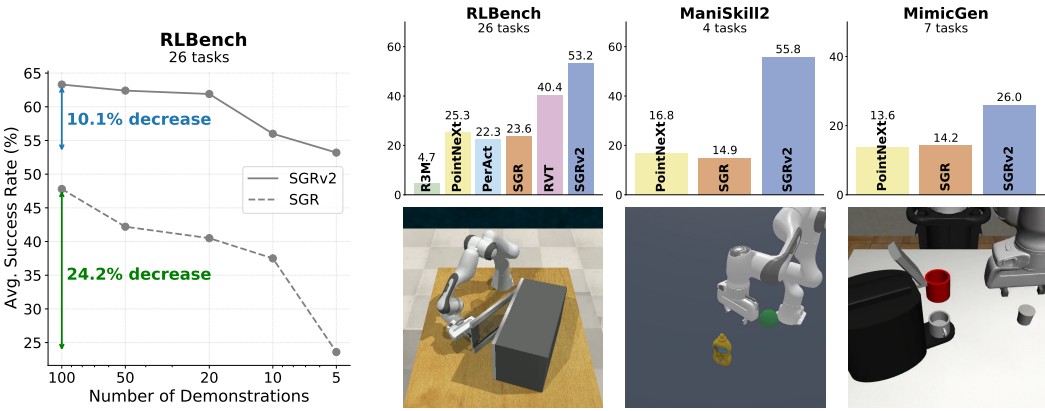

Figure 1: **Left:** Sample efficiency of SGRv2. We evaluate SGR and SGRv2 on 26 RLBench tasks, with demonstration numbers ranging from 100 to 5. Results indicate that, owing to the locality of SGRv2, it exhibits exceptional sample efficiency, with its success rate declining by only about 10%. **Top Right:** Overview of simulation results. We test SGRv2 on 3 benchmarks, consistently outperforming the baselines. **Bottom Right:** Tasks of the 3 simulation benchmarks.

To effectively utilize the inductive bias of action locality, we introduce SGRv2, a systematic framework of visuomotor policy that considers both visual and action representations. As shown in Figure 2, SGRv2 builds on the foundation of SGR but integrates action locality throughout the entire framework. SGRv2 demonstrates exceptional sample efficiency and consistently outperforms its predecessor across various demonstration sizes, achieving remarkable results with as few as 5 demonstrations, compared to SGR's performance with 100 demonstrations. The key algorithmic designs that lead to this achievement include: (i) an encoder-decoder architecture for extracting point-wise features, (ii) a strategy for predicting the *relative* target position to ensure translation equivariance, (iii) the application of point-wise weights to highlight critical local regions, and (iv) dense supervision to enhance learning efficiency.

We conduct an extensive evaluation of SGRv2 through behavior cloning across three benchmarks: RLBench [12], where keyframe control is utilized, and ManiSkill2 [13] and MimicGen [14], where dense control is applied (refer to Section 3.1 for an discussion on keyframe versus dense control). SGRv2 significantly surpasses both SGR [10] and PointNeXt [15] across these benchmarks and consistently outperforms baselines, including R3M [9], PerAct [16], and RVT [17] on RLBench. To confirm the necessity of our locality design, we conduct a series of ablation studies. Additionally, real-world experiments with Franka Emika Panda robot demonstrate SGRv2's capability to complete complex long-horizon tasks across 10 sub-tasks, validating its effectiveness. Further experiments on real-world generalization underscore SGRv2's remarkable ability to generalize.

## 2 Related Work

**Semantic Representation Learning for Robotics.** Numerous studies have focused on learning visual representations from *in-domain* data, tailored specifically to the relevant environment and task [18, 19, 20, 21, 22, 23, 24]. However, the efficacy of these methods is limited by the availability of robot data. Consequently, various efforts have been made to pre-train on large-scale *out-of-domain* data for visuo-motor control [25, 26, 27, 8, 28, 29, 30, 31, 32]. Notably, R3M [9] has demonstrated that models pre-trained on human video data can serve as a frozen perception module, facilitating downstream policy learning. Nonetheless, these approaches predominantly prioritize the pre-training of 2D visual representations, thus overlooking critical 3D geometric information, which is essential for enhancing spatial manipulation skills in robotics.

**3D Representation Learning for Robotics.** Recent works have increasingly explored the 3D representations in robotics. Studies such as C2F-ARM [33], PerAct [16], and GNFactor [34] employ voxelized observations to derive representations. However, the creation and reasoning over voxels entail a high computational burden. In contrast, RVT [17] and some researches [35, 36, 37] leverage projection techniques to generate multi-view images, extracting representation in 2D spaces

and thereby reducing computational demands. Nevertheless, these methods do not incorporate 3D geometry in the process of representation extraction, consequently limiting their capacity for 3D spatial reasoning. Point-based models, such as PointNet++ [38] and PointNeXt [15], efficiently conserve computational resources while directly processing 3D information. These models serve as the foundation for numerous robotics studies [39, 40, 41, 42, 43, 44, 45, 46, 47]. Specifically, SGR [10] utilizes point-based models to extract 3D geometric information and employs 2D foundational models for semantic understanding, integrating both to enrich representations for downstream tasks. However, the approach of SGR to extract actions from a global vector does not effectively harness locality information, thereby leading to suboptimal sample efficiency.

**Incorporating Inductive Biases into Robot Learning.** Incorporating inductive biases is an essential strategy for enhancing sample efficiency in neural network designs. Several studies have sought to improve sample efficiency by designing networks that adhere to *equivariance* properties. For example, Act3D [45] employs a translation-equivariant neural network architecture, while USEEK [48] utilizes an SE(3)-equivariant keypoint detector for one-shot imitation learning. However, these approaches are based on global equivariance [49], which may not be effective when there are relative movements between the object and the environment, which is prevalent in robot manipulation. To tackle this issue, NDFs [50] and EquivAct [51] segment the manipulated object before utilizing the equivariant models. However, given their reliance on a well-segmented point cloud, these methods are ineffective when the object is required to interact with its surroundings.

Some studies integrate *locality* into their models. L-NDF [52] incorporates locality through voxel partitioning, EDFs [49] and RiEMann [53] achieves both equivariance and locality via local message-passing mechanisms in SE(3)-Transformers [54]. Nonetheless, these methods require some hyperparameters to specify a proper size of receptive field, which limits their flexibility and extensibility. It also incurs a tradeoff between locality and expressiveness. Another line of work, such as Transporter [55], PerAct [16], and RVT [17], integrates locality by modeling action as the maximizer of scores on a predefined grid of locations. However, these designs render a voluminous action representation and suffer from quantization artifacts [56]. In contrast, our approach not only satisfies translation equivariance without relying on highly non-local centroid subtraction [50, 57] but also adaptively determines the local scope required for the task through a learnable weight, without sacrificing expressiveness or introducing a grid.

## 3 Method

In this section, we present a detailed description of the methodologies employed in SGRv2. Initially, we present an overview of SGR, keyframe and dense control, along with the problem formulation, as outlined in Section 3.1. Subsequently, we explore how to leverage the inductive bias of locality to enhance sample efficiency in robotic manipulation learning, as discussed in Section 3.2. Finally, we describe the training approaches under different control modes, detailed in Section 3.3.

### 3.1 Background

**Semantic-Geometric Representation (SGR) [10].** SGR is composed of the semantic branch, geometric branch and fusion network. In the semantic branch, the RGB part of RGB-D images are fed into a CLIP [58] image encoder, and the resulting semantic features are then back-projected onto the 3D points. The geometric branch and fusion network split a PointNeXt encoder into two stages, with semantic features injected at the interface between them. SGR models the action by solely relying on the global features extracted by the PointNeXt [15] encoder, without utilizing locality. Refer to Appendix B.2 for more details.

**Keyframe and Dense Control.** In the field of robotic manipulation, keyframe control [33, 16, 59] and dense control [13, 60] are two prevalent control modes. Keyframe control outputs a few sparse target poses, which are then executed through a motion planner. It exhibits reduced compounding errors [59], and enhanced suitability for visuomotor tasks that leverage visual priors [33, 16, 17] at the cost of inferior flexibility and expressiveness. Dense control, on the other hand, generates a

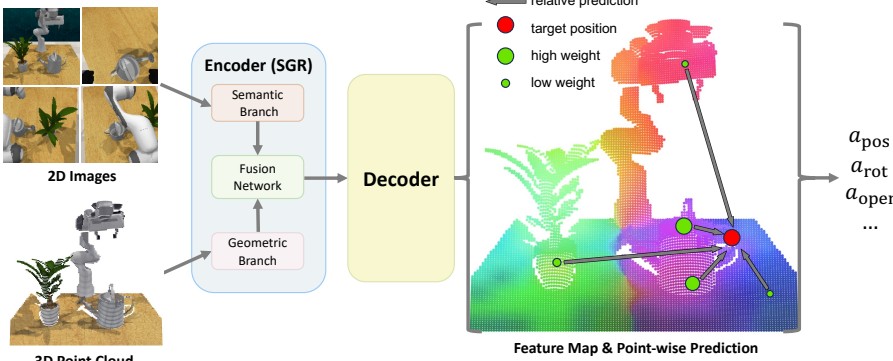

Figure 2: **SGRv2 Architecture.** Built upon SGR, SGRv2 integrates locality into its framework through four primary designs: an encoder-decoder architecture for point-wise features, a strategy for predicting relative target position, a weighted average for focusing on critical local regions, and a dense supervision strategy (not shown in the figure). This illustration specifically represents the `water plants` task. For simplicity in the visualization, we omit the proprioceptive data that is concatenated with the RGB of the point cloud before being fed into the geometric branch.

dense sequence with hundreds of actions to control the robot directly. It is applicable across a wide range of robotic scenarios but faces significant challenges with compounding errors. Given that the two models are complementary, it is highly compelling to construct a framework that supports both simultaneously. However, previous research has predominantly focused on a single control mode [16, 17, 61]. In contrast, our framework can support both modes seamlessly.

**Problem Formulation.** We frame our task as a vision-based robot manipulation problem. At each timestep, the robot receives an observation $O$ comprising single or multi-view RGB-D images $\{I_k\}_{k=1}^{K}$ and proprioceptive data $z$. For keyframe control, following the setup in PerAct [16], an action consists of the position, rotation, gripper open state, and collision indicator: $a^{\text{keyframe}} = \{a_{\text{pos}}, a_{\text{rot}}, a_{\text{open}}, a_{\text{collide}}\}$. For dense actions, as illustrated in ManiSkill2 [13] and robosuite[60], an action consists of the delta position, delta rotation, and gripper open state [13, 60, 14]: $a^{\text{dense}} = \{a_{\Delta\text{pos}}, a_{\Delta\text{rot}}, a_{\text{open}}\}$. We assume we are given $N$ expert demonstration trajectories $D = \{\tau_i\}_{i=1}^{N}$. Each trajectory $\tau_i$ is a sequence of observation-action pairs $(o_1, a_1, \ldots, a_{T-1}, o_T)$. The robot is then trained using the Behavioral Cloning (BC) algorithm with these demonstrations.

## 3.2 Locality Aware Action Modeling

To develop a sample-efficient framework for robotic manipulation that is effective in both keyframe and dense control scenarios, we capitalize on the inductive bias that actions exhibit locality properties and build our locality aware action modeling on the top of SGR. In SGRv2, we achieve locality through 4 primary designs: (1) an encoder-decoder architecture, (2) a strategy for predicting point-wise relative position formulation with (3) a learned weight, and (4) a dense supervision strategy. Refer to Figure 2 for an overview of our designs, Table 3 for ablation studies, and Appendix B.1 for architecture details.

**Encoder-Decoder Architecture.** In contrast to the encoder-only architecture used by SGR [10], we employ the encoder-decoder architecture of PointNeXt [15], which is a U-Net like architecture that excels in dense prediction tasks (e.g. segmentation). This architecture can yield a feature enriched with both global and local information for each point, namely $f_i \in \mathbb{R}^C$ for the $i$-th point, where $C$ is the dimension of the feature. Note that as designed in PointNeXt [15], the output features are solely dependent on *relative* coordinates, ensuring that the point-wise features $f_i$ remain invariant to translational transformations of the input coordinates. This point-wise features serve as the cornerstone of our locality aware action modeling.

**Relative Position Predictions.** With the point-wise features, we can predict an action at each point. Our key insight is that the end-effector usually moves towards a target close to a specific object within each execution stage. Thus, it is natural to predict the displacement of the target relative to each point. Driven by this insight, for *keyframe*, we represent the position component of a *keyframe* action $a_{\text{pos}}$ by $p_i + \Delta p(f_i)$ for the $i$-th point, where $p_i$ and $f_i$ are the coordinate and point-wise feature of the $i$-th point respectively, and $\Delta p$ is a Multilayer Perceptrons (MLP). For *dense* control, we can

predict the delta position component $a_{\Delta\text{pos}}$ of a *dense* action by modeling its direction (towards a target) and magnitude separately. Concretely, we predict the direction $\frac{a_{\Delta\text{pos}}}{||a_{\Delta\text{pos}}||_2}$ by $\frac{p_i+\Delta p(f_i)}{||p_i+\Delta p(f_i)||_2}$ and the magnitude $||a_{\Delta\text{pos}}||_2$ by $m(f_i)$, where $m$ is another MLP. In dense control, given that we employ the end-effector coordinate frame for the point cloud input[2], $p_i + \Delta p(f_i)$ can be interpreted as the target position relative to the end-effector. Consequently, $\frac{p_i+\Delta p(f_i)}{||p_i+\Delta p(f_i)||_2}$ can represent the direction of movement. The utilization of relative position leverages the locality information and achieves translation equivariance without depending on the extensive non-local centroid subtraction [50, 57]. This greatly aids in enhancing sample efficiency.

For other action components, we directly predict rotation by $r(f_i)$, gripper open state by $o(f_i)$ and collision indicator by $c(f_i)$, where $r, o, c$ are MLPs.

**Weighted Average Actions.** After obtaining the point-wise action predictions, we need to integrate these predictions into an aggregated action prediction. We adopt a simple yet effective strategy, namely weighted average. For each component, including position (which is broken down into direction and magnitude in dense control), rotation, gripper open state, and collision indicators, we employ a learned weight $w_*(f_i)$ for each point. Here, $w_*$ denotes separate MLPs (softmax is applied for normalization) for each component. The motivation behind this design is that only a few regions within the point cloud are crucial for accomplishing the task. For instance, in tasks such as picking up a cube, points located on the cube itself are more informative than those on the surrounding table. By learning these weights, we enable the aggregated prediction to concentrate on the most predictive local regions, thereby enhancing both the overall accuracy and sample efficiency.

**Dense Supervision.** To enhance the learning efficiency of local features, we adopt a dense supervision strategy. This approach integrates both aggregated action predictions and point-wise action predictions into the loss function, expressed as $\mathcal{L}_* = \mathcal{L}_*^{\text{aggregated}} + \mathcal{L}_*^{\text{points}}$. To compute $\mathcal{L}_*^{\text{aggregated}}$ and $\mathcal{L}_*^{\text{points}}$, we adopt the same loss formulation with the same ground-truth labels. Dense supervision provides feedback for all points, enabling models to learn local features more efficiently.

### 3.3 Training

**Smoothness Regularization for Dense Control.** As illustrated in point-wise relative predictions, we predict the direction and the magnitude of the delta position component $a_{\Delta\text{pos}}$ separately. Thus, the position loss is accordingly decomposed into $\mathcal{L}_{\text{dir}}$ and $\mathcal{L}_{\text{mag}}$, i.e. $\mathcal{L}_{\text{pos}} = \mathcal{L}_{\text{dir}} + \mathcal{L}_{\text{mag}}$. We predict the direction by $\frac{p_i+\Delta p(f_i)}{||p_i+\Delta p(f_i)||_2}$, which poses an underdetermined problem and allows $\Delta p$ to output spuriously large values. To mitigate this issue, we incorporate a smoothness regularization loss $\mathcal{L}_{\text{reg}} = \frac{1}{N^2} \sum_{i,j} ||(p_i + \Delta p(f_i)) - (p_j + \Delta p(f_j))||_2^2$. This loss enforces the consistency between $p_i + \Delta p(f_i)$ and $p_j + \Delta p(f_j)$ for any two points $i$ and $j$.

**Overview of Losses.** In keyframe control, following SGR [10], position is represented by a continuous 3D vector: $a_{\text{pos}} \in \mathbb{R}^3$. For rotations, the ground-truth action is represented as a one-hot vector per rotation axis with $R$ rotation bins: $a_{\text{rot}} \in \mathbb{R}^{(360/R)\times 3}$ ($R = 5$ degrees in our implementation). Open and collide actions are binary one-hot vectors: $a_{\text{open}} \in \mathbb{R}^2$, $a_{\text{collide}} \in \mathbb{R}^2$. In this way, our loss objective is as follows:

$$\mathcal{L}_{\text{keyframe}} = \alpha_1 \mathcal{L}_{\text{pos}} + \alpha_2 \mathcal{L}_{\text{rot}} + \alpha_3 \mathcal{L}_{\text{open}} + \alpha_4 \mathcal{L}_{\text{collide}}, \tag{1}$$

where $\mathcal{L}_{\text{pos}}$ is L1 loss, and $\mathcal{L}_{\text{rot}}$, $\mathcal{L}_{\text{open}}$, and $\mathcal{L}_{\text{collide}}$ are cross-entropy losses.

In dense control, following ManiSkill2 [13] and robosuite [60], both delta position and delta rotation (in the form of axis-angle coordinates) are represented by continuous 3D vectors: $a_{\Delta\text{pos}} \in \mathbb{R}^3$, $a_{\Delta\text{rot}} \in \mathbb{R}^3$, while open actions are still binary one-hot vectors: $a_{\text{open}} \in \mathbb{R}^2$. Our loss objective is:

$$\mathcal{L}_{\text{dense}} = \beta_1 (\mathcal{L}_{\text{dir}} + \mathcal{L}_{\text{mag}}) + \beta_2 \mathcal{L}_{\text{rot}} + \beta_3 \mathcal{L}_{\text{open}} + \beta_4 \mathcal{L}_{\text{reg}}, \tag{2}$$

where $\mathcal{L}_{\text{dir}}$, $\mathcal{L}_{\text{mag}}$ and $\mathcal{L}_{\text{rot}}$ are MSE losses, $\mathcal{L}_{\text{open}}$ is cross-entropy loss, and $\mathcal{L}_{\text{reg}}$ is smoothness regularization loss. For more training details, we refer to Appendix B.3.

---

[2]Motivated by FrameMiner [44], in dense control, we transform the point coordinates into the end-effector frame, positioning the end-effector at the origin of coordinates for easier computation and performance benefit.

| Method | Avg. Success ↑ | Avg. Rank ↓ | Open Microwave | Open Door | Water Plants | Toilet Seat Up | Phone On Base | Put Books | Take Out Umbrella | Open Fridge | Open Drawer | Slide Block | Sweep To Dustpan | Meat Off Grill |
|---|---|---|---|---|---|---|---|---|---|---|---|---|---|---|
| R3M | 4.7 | 5.8 | 0.9 | 36.4 | 2.9 | 15.5 | 0.0 | 0.5 | 5.2 | 3.2 | 0.0 | 24.0 | 0.4 | 0.1 |
| PointNeXt | 25.3 | 3.4 | 7.1 | 60.9 | 5.6 | 49.9 | 46.4 | 57.5 | 37.5 | 9.2 | 21.7 | 59.5 | 42.0 | 59.9 |
| PerAct | 22.3 | 4.1 | 4.3 | 59.6 | 28.5 | 69.3 | 0.0 | 25.1 | 75.9 | 3.1 | 56.4 | 47.5 | 2.8 | 85.9 |
| SGR | 23.6 | 4.1 | 6.4 | 55.3 | 24.9 | 30.7 | 47.2 | 29.3 | 36.3 | 7.1 | 31.9 | 72.0 | 43.6 | 52.7 |
| RVT | 40.4 | 2.2 | 18.3 | 71.2 | 34.8 | 47.6 | 62.3 | 46.5 | **85.3** | **24.0** | 75.1 | 85.1 | 19.6 | 90.5 |
| SGRv2 (ours) | **53.2** | **1.2** | **27.2** | **76.8** | **38.0** | **89.6** | **84.1** | **63.7** | 74.5 | 13.2 | **81.3** | **100.0** | **61.5** | **96.5** |

| Method | Turn Tap | Put In Drawer | Close Jar | Drag Stick | Stack Blocks | Screw Bulb | Put In Safe | Place Wine | Put In Cupboard | Sort Shape | Push Buttons | Insert Peg | Stack Cups | Place Cups |
|---|---|---|---|---|---|---|---|---|---|---|---|---|---|---|
| R3M | 26.1 | 0.0 | 0.0 | 0.3 | 0.0 | 0.0 | 0.3 | 0.4 | 0.0 | 0.0 | 6.8 | 0.0 | 0.0 | 0.0 |
| PointNeXt | 48.7 | 17.1 | **36.0** | 18.5 | 1.9 | 4.1 | 12.1 | 31.5 | 3.3 | 0.4 | 22.0 | 0.1 | 4.4 | 0.4 |
| PerAct | 8.0 | 0.1 | 0.5 | 10.3 | 1.7 | 4.4 | 0.9 | 8.7 | 0.4 | 0.4 | 83.1 | 1.9 | 0.1 | 0.7 |
| SGR | 34.4 | 8.3 | 13.3 | 64.4 | 0.0 | 0.9 | 16.9 | 24.7 | 0.1 | 0.1 | 12.0 | 0.1 | 0.0 | 1.1 |
| RVT | 38.4 | 19.6 | 25.2 | 45.7 | 8.8 | 24.0 | 30.7 | **92.7** | 5.6 | 1.6 | 90.4 | 4.0 | 3.1 | 1.2 |
| SGRv2 (ours) | **87.9** | **75.9** | 25.6 | **94.9** | **17.5** | **24.1** | **55.6** | 53.1 | **20.3** | **1.9** | **93.2** | **4.1** | **21.3** | **1.6** |

Table 1: **Performance on RLBench with 5 Demonstrations.** All numbers represent percentage success rates averaged over 3 seeds. See Appendix F for standard deviation. SGRv2 outperforms the most competitive baseline RVT on 23/26 tasks, with an average improvement of 1.32×.

## 4 Experiments

Our experiments are designed to answer the following questions: (1) How does SGRv2 perform when locality is incorporated into designs, especially in data-limited scenarios, compared to various 2D and 3D representations? (2) Can SGRv2 consistently demonstrate advantages across different control modes? (3) What are the contributions of the key components of SGRv2's locality design to its overall performance? (4) How does SGRv2 perform in real-robot tasks, and does it possess the ability to generalize in the real world?

### 4.1 Simulation Setup

**Environment and Tasks.** The simulation experiments are conducted on 3 robot learning benchmarks: RLBench [12], ManiSkill2 [13], and MimicGen [14]. RLBench is a large-scale benchmark designed for vision-guided manipulation. Following previous works [33, 16, 17] we use keyframe control on RLBench. ManiSkill2 is a comprehensive benchmark for manipulation skills, enhancing diversity with object-level variations. MimicGen generates large-scale robot learning datasets from human demonstrations. On ManiSkill2 and MimicGen, following prior works [61, 14], we use dense control. On RLBench, we use 4 RGB-D cameras positioned at the front, left shoulder, right shoulder, and wrist of a Franka Emika Panda, while on ManiSkill2 and MimicGen, we use a front-view and a wrist-view RGB-D camera. We use 26 RLBench tasks with 5 demonstrations per task, and 4 ManiSkill2 tasks and 7 MimicGen tasks with 50 demonstrations per task (except for *PickSingleYCB*, where 50 demonstrations per object are used). On MimicGen, we use $D_1$ initial distribution, which presents a broader and more challenging range. See Appendix A for more details.

**Evaluation.** The evaluation approach is designed to minimize variance in our results. On RLBench, we train an agent for 20,000 iterations and save checkpoints every 800 iterations, while in ManiSkill2 and MimicGen, we train for 100,000 iterations and save checkpoints every 4,000 iterations. Then we evaluate the last 5 checkpoints for 50 episodes and get the average success rates [3]. Finally, we conduct the experiments with 3 seeds and report the average results.

**Baselines.** We compare SGRv2 against the following baselines: (1) **R3M** [9] is a 2D visual representation designed for robotic manipulation, which is pre-trained on large-scale human video datasets. For fair comparisons, we utilize frozen R3M to process RGB images, employ a separate 2D CNN to process depth images, and subsequently fuse the two resulting features. (2) **PointNeXt** [15] is an enhanced version of the classic PointNet++ architecture for point cloud processing. We employ the encoder of PointNeXt to obtain the 3D representation. (3) **PerAct** [16] is a 3D representation that voxelizes the workspace and utilizes a Perceiver Transformer [62] to process voxelized observations. (4) **SGR** [10] is a representation that integrates both high-level 2D semantic understanding and low-level 3D spatial reasoning. (5) **RVT** [17] is a 3D representation that utilizes a multi-view transformer to predict actions and integrates these predictions into a 3D space through back-projection from multiple viewpoints.

---

[3]MimicGen [14] reports *maximum* results across different checkpoints, while we provide *average* results, offering a more robust and realistic measure of model performance.

| Method | Avg. Success ↑ | Avg. Rank ↓ | LiftCube | PickCube | StackCube | PickSingleYCB |
|---|---|---|---|---|---|---|
| PointNeXt | 16.8 | 2.5 | 50.8 | 4.7 | 10.6 | 1.1 |
| SGR | 14.9 | 2.5 | 26.9 | 12.2 | 3.5 | 17.0 |
| SGRv2 (ours) | **55.8** | **1.0** | **80.5** | **72.9** | **27.7** | **42.2** |

| Method | Avg. Success ↑ | Avg. Rank ↓ | Stack | StackThree | Square | Threading | Coffee | HammerCleanup | MugCleanup |
|---|---|---|---|---|---|---|---|---|---|
| PointNeXt | 13.6 | 2.9 | 56.1 | 3.7 | 0.9 | 3.6 | 12.0 | 11.7 | 7.1 |
| SGR | 14.2 | 2.0 | 50.8 | 5.6 | 1.3 | 4.0 | 14.1 | 14.1 | **9.7** |
| SGRv2 (ours) | **26.0** | **1.0** | **81.2** | **37.9** | **2.8** | **6.7** | **27.9** | **16.1** | **9.7** |

Table 2: **Performance on ManiSkill2 (top) and MimicGen (bottom) with 50 Demonstrations.** We report success rates averaged over 3 seeds. See Appendix F for standard deviation. We observe that SGRv2 consistently outperforms baselines like SGR and PointNeXt.

## 4.2 Simulation Results

**RLBench Performance.** Figure 1 depicts the performance of SGR and SGRv2 across differing number of demonstrations. Initially, as the availability of data decreases, we note a significant decline in SGR's performance compared to SGRv2, highlighting the difficulties in maintaining model performances without the inductive bias towards locality awareness. Additionally, Table 1 provides a comparison of success rates obtained with only 5 demonstrations using various representations. We observe that the absence of 3D geometric information, as demonstrated by R3M using a 2D representation, results in markedly low performance, highlighting the critical role of 3D priors in robotic manipulation under data-constrained scenarios. Finally, SGRv2 is demonstrated to be the superior representation, achieving 53.2% average success rate with merely 5 demonstrations and significantly outperforming the most competitive baseline, RVT, with an average improvement factor of $1.32\times$ and achieving enhanced performance in 23 out of 26 tasks. The results underscore the advantages of incorporating 3D geometry and well-designed locality to enhance sample efficiency.

**ManiSkill2 and MimicGen Performance.** To evaluate the performance of SGRv2 in dense control scenarios, we conduct comparisons with several baselines on ManiSkill2 and MimicGen. Our experimental results, summarized in Table 2, demonstrate that SGRv2 significantly outperforms the baselines. In particular, thanks to our tailored approach for dense control, SGRv2 exhibits superior performance in tasks where the object's location consistently aligns with the direction of the delta actions, such as `Pick Cube` and `Stack Three`. These findings confirm that SGRv2 acts as a sample-efficient, universal representation adept at handling both keyframe and dense control scenarios. See Appendix D for results on different numbers of demonstrations and additional baselines.

**Ablations.** As shown in Table 3, we conduct ablations to assess the locality design choices of SGRv2. (1) Decoder architecture is the cornerstone of our locality designs. Omitting the decoder from the SGRv2 would prevent the application of other locality designs, forcing us to rely solely on the global representation from the encoder. This would lead to a significant decrease in performance. (2) Predicting absolute positions, results in markedly poorer performance. This underscores that relative position

| Method | Avg. success |
|---|---|
| SGRv2 | **53.2** |
| SGRv2 w/o decoder | 21.3 |
| SGRv2 w/ absolute pos prediction | 21.0 |
| SGRv2 w/ uniform point weight | 44.2 |
| SGRv2 w/o dense supervision | 40.1 |

Table 3: **Ablations.** Average success rate of 26 RLBench tasks with 5 demonstrations after ablating key components of locality design.

predictions are the key insight of the locality design. (3) Substituting point-wise weights with uniform weights reduces performance, confirming the role of point-wise weighting in focusing on predictive local regions. (4) Eliminating dense supervision leads to a decline in overall performance, illustrating that dense supervision enhances the model's learning efficacy.

**Emergent Capabilities.** In our study, we visualize the point-wise weights of SGRv2 detailed in Section 3.2. As depicted in Figure 3, we sequentially visualize point clouds with RGB and point-wise weights. Surprisingly, the results consistently demonstrate an alignment between points with higher weights (in red) and the object affordances, which denote the functional areas of objects. This observation highlights the capability of SGRv2 to precisely identify and emphasize critical local regions on objects. Motivated by this, we conduct experiments on visual distractors in Appendix E.

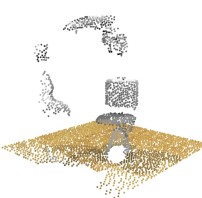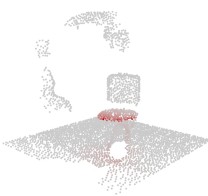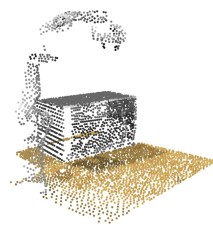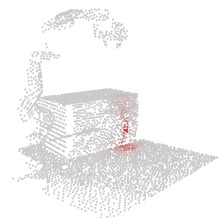

Figure 3: **Emergent Capabilities.** We visualize the point-specific weights and find that the points with high weights (in red) align with the object's affordances. Left: `toilet seat up`. Right: `open microwave`.

| Task | Sub-task | PerAct | RVT | SGRv2 |
|---|---|---|---|---|
| Tidy Up the Table | Put trash in trash can | 50 | 50 | 80 |
| | Put socks in box | 60 | 80 | 90 |
| | Put marker in pen holder | 10 | 10 | 30 |
| | Open drawer | 20 | 40 | 60 |
| | Put lollipop in drawer | 10 | 10 | 30 |
| | Close drawer | 40 | 60 | 80 |
| Make Coffee | Turn on coffee machine | 100 | 100 | 100 |
| | Put funnel onto carafe | 0 | 20 | 80 |
| | Pour powder into funnel | 0 | 10 | 10 |
| | Pour water | 10 | 30 | 70 |
| Avg. Success Rate | | 30 | 41 | **63** |

Figure 4: **Left:** Real-robot long-horizon tasks. **Right:** Success rate (%) of multi-task agents on real-robot tasks. We collect 8 demonstrations and evaluate 10 episodes for each task.

## 4.3   Real-Robot Results

To evaluate the effectiveness of SGRv2 in a real-robot setting, we conduct experiments using the Franka Emika Panda across two long-horizon tasks (a total of 10 sub-tasks), and a generalization task. Refer to Appendix C for more details on robot setup, task designs and failure cases discussions.

**Long-horizon Tasks.** We collect 8 demonstrations per task and train multi-task agents. Each sub-task is tested across 10 episodes. We present comparative results between PerAct, RVT, and SGRv2 in Figure 4, where SGRv2 demonstrates a substantial performance advantage. A common issue with PerAct is that it tends to bias towards occupied voxels, frequently causing collisions. For RVT, typical failures stem from large position errors along the camera's viewing direction, potentially caused from the inaccuracies in the virtual images rendered orthogonal to the real camera's perspective.

**Generalization Task.** We assess the generalization capability of SGRv2 by employing 6 cups of different colors. Each scene involves one target cup and two distractor cups of different colors. We collect 5 demonstrations for each of the 4 colors and test the model on both 4 seen and 2 unseen scenarios. Each color is tested across 10 episodes. As indicated

| Method | Seen | Unseen |
|---|---|---|
| PerAct | 100 | 10 |
| RVT | 100 | 5 |
| SGRv2 w/o sem. | 100 | 0 |
| SGRv2 | 100 | **70** |

in the right table, compared with PerAct and RVT, SGRv2 exhibits the ability to generalize across color variations. The results show SGRv2 without the semantic branch achieves zero performance on unseen colors, underlining the critical role of semantic awareness in enhancing generalization.

## 5   Discussion

**Limitations.** Our models are currently trained using a vanilla BC objective. A promising direction involves integrating the Diffusion Policy [2], which excels in dealing with multimodal and heterogeneous trajectories, with our locality framework to further enhance performance in real world. Additionally, due to our focus on sample efficiency, the evaluations on generalization are currently insufficient. We are eager to expand this work to include a broader range of generalization aspects, such as object shapes, camera positions, and background textures.

**Conclusion.** We present SGRv2, a systematic framework of visuomotor policy that considers both visual and action representations. Built upon the SGR, SGRv2 integrates action locality across its entire framework. Through comprehensive evaluations across a diverse range of tasks in multiple simulated and real-world environments with limited data availability, SGRv2 exhibits superior performance and outstanding sample efficiency.

**Acknowledgments**

We thank Haoxu Huang and Fanqi Lin for their assistance in conducting real-robot experiments. This work is supported by the Ministry of Science and Technology of the People's Republic of China, the 2030 Innovation Megaprojects "Program on New Generation Artificial Intelligence" (Grant No. 2021AAA0150000). This work is also supported by the National Key R&D Program of China (2022ZD0161700).

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

# A  Simulation Task Details

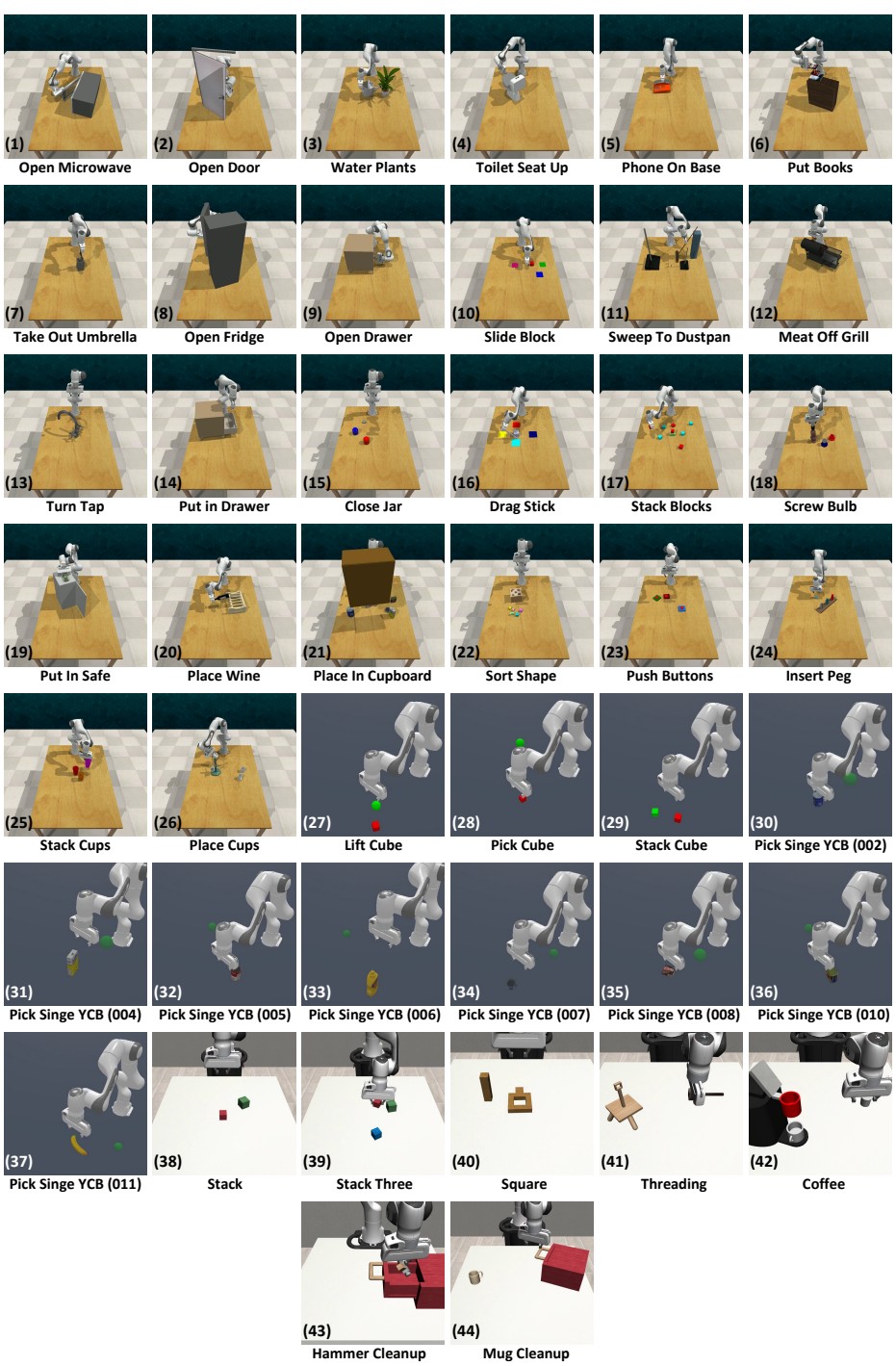

Figure 5: **Simulation Tasks.** Our simulation experiments encompass 26 tasks (1-26) from RL-Bench, 4 tasks (27-37, where 30-37 are 8 different YCB [63] objects of task `Pick Single YCB`) from ManiSkill2, and 7 tasks (38-44) from MimicGen.

Our simulation experiments are conducted on 3 robot learning benchmarks: RLBench [12], ManiSkill2 [13], and MimicGen [14]. See Figure 5 for an overview of the simulation tasks. In these

simulations, all cameras have a resolution of $128 \times 128$. In the following, we will provide a detailed examination of tasks from the three benchmarks.

## A.1 RLBench Tasks

We utilize 26 RLBench tasks, including 8 tasks used in SGR [10] and 18 tasks used in PerAct [16] and RVT [17]. For tasks with multiple variations, we use the first variation. In RLBench, we use 5 demonstrations per task, unless specified otherwise. Given that SGR and PerAct provide detailed descriptions of these RLBench tasks, we omit these details here for simplicity.

## A.2 ManiSkill2 Tasks

We utilize 4 ManiSkill2 tasks, each described in detail as follows. (1) **Lift Cube**: Pick up a red cube and lift it to a specified height. (2) **Pick Cube**: Pick up a red cube and move it to a target position. (3) **Stack Cube**: Pick up a red cube and place it onto a green cube. (4) **Pick Single YCB**: Pick up a YCB [63] object and move it to the target position. In our experiments, we use 8 YCB objects (excluding those that are too difficult to pick up): *002_master_chef_can*, *004_sugar_box*, *005_tomato_soup_can*, *006_mustard_bottle*, *007_tuna_fish_can*, *008_pudding_box*, *010_potted_meat_can*, *011_banana*. For the first three tasks, we utilize 50 demonstrations per task, while for the last one (Pick Single YCB), we employ 50 demonstrations per YCB object.

## A.3 MimicGen Tasks

We utilize 7 MimicGen tasks with 50 demonstrations per task, all employing the initial distribution $D_1$, which presents a broader and more challenging range. The details are as follows: (1) **Stack**: Stack a red block on a green one. (2) **Stack Three**: Similar to Stack, but with an additional step of stacking a blue block on the red one. (3) **Square**: Pick up a square nut and place it on a peg. (4) **Threading**: Pick up a needle and thread it through a hole in a tripod. (5) **Coffee**: Pick up a coffee pod, insert it into the coffee machine, and close the machine hinge. (6) **Hammer Cleanup**: Open a drawer, pick up a hammer, place it back into the drawer, and close the drawer. (7) **Mug Cleanup**: Similar to Hammer Cleanup, but with a mug.

# B  SGRv2 Details

## B.1  Architecture Details

**Input Data.** The SGRv2 model takes as input RGB images $\{I_k\}_{k=1}^{K}$ of size $H \times W$ and corresponding depth images of the same size from multiple camera views. Point clouds are generated from these depth images using known camera extrinsics and intrinsics. A crucial aspect is the alignment of the RGB images with the point clouds, ensuring a precise one-to-one correspondence between elements in the two data forms. For keyframe control, the point cloud is represented in the robot's base frame. In contrast, for dense control—inspired by FrameMiner [44]—the point cloud is transformed into the end-effector frame to simplify computation and enhance performance.

For keyframe control, the model additionally receives proprioceptive data $z$, which includes four scalar values: gripper open state, left finger joint position, right finger joint position, and action sequence timestep. In dense control, proprioceptive data is not utilized. Additionally, following SGR [10], if a task comes with language instruction $S$, this also forms part of the model's input.

**Other Details.** Following SGR [10], we use CLIP-ResNet-50 as the image encoder for the semantic branch. For the 3D encoder-decoder, we employ PointNeXt-XL. The output from the encoder-decoder is a point-wise feature, denoted as $f_i^{\text{raw}} \in \mathbb{R}^C$ for the $i$-th point, where the feature dimension $C$ is 64. We apply a linear layer followed by a ReLU activation to produce a processed point-wise feature $f_i$, increasing the feature dimension to 256. We then predict the relative position $\Delta p(f_i)$, magnitude $m(f_i)$ (for dense control), rotation $r(f_i)$, gripper open state $o(f_i)$, and collision indicator $c(f_i)$ (for keyframe control) of the $i$-th point, where $\Delta p, m, r, o, c$ are 3-layer MLPs. Note that when

representing the ground-truth actions as one-hot vectors—such as rotation, gripper open state, and collision indicators in keyframe control—the action predictions correspond to the output probabilities following the softmax layer. Finally, for each action component, we assign a learned weight $w_*(f_i)$ to each point, where $w_*$ represents separate 3-layer MLPs with softmax normalization across the points dimension.

## B.2  SGR Details

SGRv2 is built upon SGR [10], which we briefly introduced in Section 3.1. Here, we provide a detailed description of SGR's three components: semantic branch, geometric branch, and fusion network.

**Semantic Branch.** Using a collection of RGB images $\{I_k\}_{k=1}^K$ from $K$ calibrated cameras, they initially apply a frozen pre-trained 2D model $\mathcal{G}$, such as CLIP's visual encoder, to extract multi-view image features $\{\mathcal{G}(I_k)\}_{k=1}^K$. When a language instruction $S$ accompanies a task, they utilize a pre-trained language model $\mathcal{H}$, like CLIP's language encoder, to generate the language features $\mathcal{H}(S)$. They align these image features $\mathcal{G}(I_k)$ with the language features $\mathcal{H}(S)$ using a visual grounding module, producing $\{M_k\}_{k=1}^K$. Subsequently, they rescale the visual or aligned feature maps to the dimensions of the original images through bilinear interpolation and reduce their channels by $1 \times 1$ convolution, generating a set of features $\{F_k\}_{k=1}^K$, where each $F_k \in \mathbb{R}^{H \times W \times C_1}$. These high-level semantic features are then back-projected into 3D space to form point-wise features for the point cloud, expressed as $F_{\text{sem}} \in \mathbb{R}^{N \times C_1}$, where $N = K \times H \times W$.

**Geometric Branch.** They construct the initial point cloud coordinates $P = \{p_i\}_{i=1}^N \in \mathbb{R}^{N \times 3}$ and RGB features $F_c \in \mathbb{R}^{N \times 3}$ using multi-view RGB-D images and camera parameters (i.e., camera intrinsics and extrinsics). Optionally, they append a $D$-dimensional vector, derived from robot proprioceptive data $z$ via a linear layer, to each point feature. They then process the point cloud coordinates $P$ and features $F_c$ through a hierarchical PointNeXt encoder, extracting compact geometric coordinates $P' \in \mathbb{R}^{M \times 3}$ and features $F'_c \in \mathbb{R}^{M \times C_2}$ ($M < N$).

**Fusion Network.** To merge the two complementary branches, they first subsample the point-wise semantic features $F_{\text{sem}}$ using the same point subsampling procedure as in the geometric branch, resulting in $F'_{\text{sem}} \in \mathbb{R}^{M \times C_1}$. They then perform a channel-wise concatenation of the semantic and geometric features to form $F_{\text{fuse}} = \text{Concat}(F'_{\text{sem}}, F'_c) \in \mathbb{R}^{M \times (C_1 + C_2)}$. Finally, the fused features are processed through several set abstraction blocks [38, 15], enabling a cohesive modeling of the cross-modal interaction between 2D semantics and 3D geometric information.

## B.3  Training Details

**Losses.** As illustrated in Section 3.3, in keyframe control, our loss objective is as follows:

$$\mathcal{L}_{\text{keyframe}} = \alpha_1 \mathcal{L}_{\text{pos}} + \alpha_2 \mathcal{L}_{\text{rot}} + \alpha_3 \mathcal{L}_{\text{open}} + \alpha_4 \mathcal{L}_{\text{collide}}, \tag{3}$$

where $\mathcal{L}_{\text{pos}}$ is L1 loss, and $\mathcal{L}_{\text{rot}}$, $\mathcal{L}_{\text{open}}$, and $\mathcal{L}_{\text{collide}}$ are cross-entropy losses. In our experiments, we set $\alpha_1 = 300$ and $\alpha_2 = \alpha_3 = \alpha_4 = 1$.

In dense control, our loss objective is:

$$\mathcal{L}_{\text{dense}} = \beta_1 (\mathcal{L}_{\text{dir}} + \mathcal{L}_{\text{mag}}) + \beta_2 \mathcal{L}_{\text{rot}} + \beta_3 \mathcal{L}_{\text{open}} + \beta_4 \mathcal{L}_{\text{reg}}, \tag{4}$$

where $\mathcal{L}_{\text{dir}}$, $\mathcal{L}_{\text{mag}}$ and $\mathcal{L}_{\text{rot}}$ are MSE losses, $\mathcal{L}_{\text{open}}$ is cross-entropy loss, and $\mathcal{L}_{\text{reg}}$ is smoothness regularization loss. In our experiments, we set $\beta_1 = 10$, $\beta_2 = \beta_3 = 1$ and $\beta_4 = 0.3$.

**Data Augmentation.** (1) **Translation and rotation perturbations**: in keyframe control, the training samples is augmented with $\pm 0.125$ m translation perturbations and $\pm 45°$ yaw rotation perturbations. (2) **Color drop** is to randomly replace colors with zero values. This technique serves as a powerful augmentation for PointNeXt [15], leading to significant enhancements in the performance of tasks where color information is available. (3) **Feature drop**: Color drop randomly replaces colors with zero values, which results in both the RGB and semantic features becoming constant.

However, there are certain tasks where colors play a crucial role, and disregarding color information in these tasks would make them unsolvable. To address this issue, we propose *feature drop*. Specifically, this involves randomly replacing the semantic features with zero values, while keeping the RGB values unchanged. (4) **Point resampling** is a widely used technique in point cloud data processing that adjusts the density of the point cloud. It involves selecting a subset of points from the original dataset to create a new dataset with a modified density. Firstly, we filter out points outside the workspace. Then in keyframe control, we resample 4096 points from the point cloud using farthest point sampling (FPS), while in dense control, we resample 1200 points using the same method. (5) **Demo augmentation** [33] [16], used in keyframe control, captures transitions from intermediate points along a trajectory to keyframe states, rather than from the initial state to the keyframe state. This approach significantly increases the volume of the training data.

**Hyperparameters.** The configuration of hyperparameters applied in our studies are shown in Table 4. For each task, the experiments are conducted on a single NVIDIA GeForce RTX 3090 GPU.

Table 4: Hyper-parameters used in our simulation experiments.

| Config | Keyframe Control | Dense Control |
|---|---|---|
| Training iterations | $20,000$ | $100,000$ |
| Leraning rate | $0.003$ | $0.0003$ |
| Batch size | 16 | 16 |
| Optimizer | AdamW | AdamW |
| Lr Scheduler | Cosine | Cosine |
| Warmup step | 200 | 0 |
| Weight decay | $1 \times 10^{-6}$ | $1 \times 10^{-6}$ |
| Color drop | 0.4 | 0 |
| Feature drop | 0 | 0.4 |
| Number of input points | 4096 | 1200 |

## B.4 Training and Inference Speed

We test the training and inference speed of SGRv2 on a NVIDIA 3090 GPU. For keyframe control, training takes approximately 5 hours for 20k steps, with an inference speed of 10 FPS. For dense control, training requires around 7 hours for 100k steps, with an inference speed of 30 FPS.

## C  Real-Robot Details

### C.1  Real-Robot Setup

For our real-robot experiments, we use a Franka Emika Panda manipulator equipped with a parallel gripper. We utilize keyframe control, and the motion planning is executed through `MoveIt` [4]. Perception is achieved through an Intel RealSense L515 camera, positioned in front of the scene. The camera generates RGB-D images with a resolution of $1280 \times 720$. We leverage the `realsense-ros`[5] to align depth images with color images. The extrinsic calibration between the camera frame and robot base frame is carried out using the `MoveIt` calibration package.

When preprocessing the RGB-D images, we resize the $1280 \times 720$ images to $256 \times 256$ using nearest-neighbor interpolation. We choose this interpolation method instead of others, like bilinear interpolation, because the latter can introduce artifacts into the depth map, resulting in a noisy point cloud. Following these steps enables us to process RGB-D images as we do in our simulation experiments. It is essential to adjust the camera's intrinsic parameters appropriately after resizing the images. We train SGRv2 for 40,000 training steps and use the final checkpoint for evaluation.

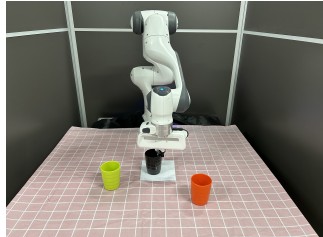

Figure 6: Real-robot generalization task.

## C.2 Real-Robot Tasks

Our real-robot experiments involve three tasks: Tidy Up the Table, Make Coffee, and Move Color Cup to Target. The first two are long-horizon tasks, while the last is a generalization task. We provide details of the task design as follows.

**Tidy Up the Table** (as shown in Figure 4 Top Left) is to place the clutter on the table in its appropriate locations. The task consists of 6 sub-tasks, each detailed as follows: (1) `Put trash in trash can`: Pick up the trash and place it in the trash can. (2) `Put socks in box`: Pick up the socks and place them in the box. (3) `Put marker in pen holder`: Pick up the marker and place it in the pen holder. (4) `Open drawer`: Grasp the drawer handle and pull it open. (5) `Put lollipop in drawer`: Pick up the lollipop and place it into the drawer. (6) `Close drawer`: Push the drawer closed.

**Make Coffee** (as shown in Figure 4 Bottom Left) is to make pour-over coffee. This task is composed of 4 sub-tasks, each described in detail as follows: (1) `Turn on coffee machine`: Press the button on the coffee machine to activate it. (2) `Put funnel onto carafe`: Pick up the funnel and place it onto the carafe. (3) `Pour powder into funnel`: Pick up the powder holder and pour the powder into the funnel. (4) `Pour water`: Pick up the kettle and pour the water onto the powder in the funnel.

**Move Color Cup to Target** (as shown in Figure 6) is to select the target color cup from three cups and move it to the white area. The target color is indicated through language instructions. We have 6 cups of different colors: *white*, *red*, *yellow*, *orange*, *black*, and *green*. Each scenario involves one target cup and two distractor cups of different colors. We collect five demonstrations for each of the first 4 colors and test the model on both 4 seen and 2 unseen scenarios.

## C.3 Discussions of Failure Cases

In the real-robot experiments, the primary failure cases of SGRv2 include: (1) For smaller objects like lollipops, the model sometimes struggles to detect them, especially when they are farther away from the camera. (2) For tasks that are sensitive to grasping position or angle, such as grasping markers or funnels, slight deviations can lead to unsuccessful attempts or cause the object to slip. (3) In keyframe control, where a few sparse target poses are output and then executed through a motion planner, failures can occur due to collisions during the motion planning phase.

# D  Additional Results

Our primary objective of Section 4 is to demonstrate the sample efficiency of SGRv2. This is why we initially choose to evaluate our method using only 5% of the data employed in the baselines' original settings. However, to make our experiments more complete, we conduct additional experiments to provide a more comprehensive evaluation on more different numbers of demonstrations and more baselines. Note that all our results are the average success rates of the last 5 checkpoints.

---

[4]https://moveit.ros.org
[5]https://github.com/IntelRealSense/realsense-ros

| Method | Avg. Success ↑ | Open Microwave | Open Door | Water Plants | Toilet Seat Up | Phone On Base | Put Books | Take Out Umbrella | Open Fridge |
|---|---|---|---|---|---|---|---|---|---|
| PointNeXt | 33.1 | 13.6 | 61.6 | 14.8 | 64.4 | 57.2 | 48.0 | 83.6 | 16.4 |
| PerAct | 36.7 | 9.2 | 78.0 | 12.0 | 83.6 | 0.0 | 18.8 | 91.6 | 14.4 |
| SGR | 47.8 | 46.0 | 76.8 | 24.4 | 59.6 | 82.8 | 92.0 | 90.0 | 26.4 |
| RVT | 52.1 | 19.2 | 79.2 | 11.2 | 62.0 | 78.4 | 63.6 | 97.2 | 18.4 |
| SGRv2 (ours) | **63.3** | 68.4 | 86.0 | 17.6 | 69.2 | 85.6 | 69.2 | 95.6 | 19.2 |

| Method | Open Drawer | Slide Block | Sweep To Dustpan | Meat Off Grill | Turn Tap | Put In Drawer | Close Jar | Drag Stick | Stack Blocks |
|---|---|---|---|---|---|---|---|---|---|
| PointNeXt | 63.6 | 83.6 | 52.4 | 0.0 | 84.8 | 1.6 | 35.6 | 0.0 | 8.8 |
| PerAct | 89.8 | 97.3 | 32.9 | 98.2 | 5.5 | 4.4 | 23.2 | 75.3 | 43.4 |
| SGR | 75.6 | 89.2 | 63.2 | 93.6 | 94.8 | 22.8 | 36.4 | 80.8 | 0.0 |
| RVT | 70.0 | 71.2 | 18.0 | 92.0 | 73.6 | 84.4 | 35.2 | 100.0 | 18.8 |
| SGRv2 (ours) | 92.8 | 94.4 | 64.4 | 97.6 | 95.2 | 80.8 | 32.4 | 94.8 | 52.0 |

| Method | Screw Bulb | Put In Safe | Place Wine | Put In Cupboard | Sort Shape | Push Buttons | Insert Peg | Stack Cups | Place Cups |
|---|---|---|---|---|---|---|---|---|---|
| PointNeXt | 21.6 | 7.2 | 13.6 | 18.0 | 2.8 | 100.0 | 1.2 | 6.0 | 0.0 |
| PerAct | 18.2 | 7.9 | 39.7 | 7.9 | 2.2 | 82.0 | 8.9 | 7.7 | 1.2 |
| SGR | 17.6 | 27.6 | 35.6 | 12.4 | 2.8 | 84.8 | 2.0 | 6.0 | 0.8 |
| RVT | 43.2 | 67.2 | 92.0 | 17.6 | 6.4 | 100.0 | 12.8 | 22.8 | 0.4 |
| SGRv2 (ours) | 68.4 | 59.2 | 68.0 | 50.4 | 6.4 | 99.2 | 8.0 | 70.4 | 0.8 |

Table 5: Performance on RLBench with 100 demonstrations.

| #Demonstrations | 100 | 50 | 20 | 10 | 5 |
|---|---|---|---|---|---|
| RVT | 52.1 | 46.3 | 43.3 | 42.3 | 40.4 |
| SGRv2 (ours) | 63.3 | 62.4 | 61.9 | 56.0 | 53.2 |

Table 6: Average performance of 26 RLBench tasks with varying number of demonstrations.

**RLBench with 100 Demonstrations.** We test SGRv2 and the baseline methods (RVT, SGR, PerAct, and PointNeXt) using 100 demonstrations. Refer to Table 5 for the results.

**RVT with Varying Demonstrations.** We test RVT (the most competitive baseline in RLBench) on 26 RLBench tasks, with demonstration numbers ranging from 100 to 5. Table 6 shows the average results of 26 tasks compared with results of SGRv2.

**MimicGen with 1000 Demonstrations.** We evaluate SGRv2 against SGR, PointNeXt, 2D BC-RNN (used in MimicGen [14] and robomimic [64]), and 2D BC (used in robomimic [64]). The latter two baselines are trained and tested using the official codes of MimicGen [6] and robomimic [7], but with different evaluation metrics-we report the average results of the last 5 checkpoints instead of the maximum of 30 checkpoints, as explained in Section 4.1. Table 7 shows the results.

**More Baselines in MimicGen with 50 Demonstrations.** In addition to the baselines included in the Table 2, we evaluated SGRv2 against the 2D BC-RNN (used in MimicGen [14] and robomimic [64]), 2D BC (used in robomimic [64]), and R3M [9] baselines with 50 demonstrations. Table 8 shows the results.

**MimicGen with 200 Demonstrations.** We further compared SGRv2 against 2D BC-RNN (used in MimicGen [14] and robomimic [64]) and 2D BC (used in robomimic [64]) using 200 demonstrations. Table 9 shows the results.

The results from these extended experiments show that SGRv2 consistently outperforms the baselines across all settings. While it's noteworthy that BC-RNN achieves comparable performance to SGRv2 when trained on 1000 demonstrations, it falls short when the number of demonstrations is reduced to 50 or 200. This highlights the superior sample efficiency of SGRv2. Additionally, we

---

[6] https://github.com/NVlabs/mimicgen
[7] https://github.com/ARISE-Initiative/robomimic

| Method | Avg. Success ↑ | Stack | StackThree | Square | Threading | Coffee | HammerCleanup | MugCleanup |
|---|---|---|---|---|---|---|---|---|
| SGR | 42.1 | 84.4 | 54.0 | 26.4 | 11.6 | 41.6 | 38.4 | 38.4 |
| PointNeXt | 42.3 | 90.4 | 72.4 | 12.4 | 12.8 | 36.4 | 33.6 | 38.0 |
| 2D BC | 32.3 | 84.4 | 54.8 | 35.6 | 13.2 | 6.8 | 26.8 | 4.8 |
| 2D BC-RNN | 63.2 | 96.0 | 74.4 | 56.8 | 34.8 | 82.8 | 46.0 | 51.6 |
| SGRv2 (ours) | **66.2** | 96.4 | 84.2 | 56.4 | 56.0 | 86.0 | 46.2 | 38.4 |

Table 7: Performance on MimicGen with 1000 demonstrations.

| Method | Avg. Success ↑ | Stack | StackThree | Square | Threading | Coffee | HammerCleanup | MugCleanup |
|---|---|---|---|---|---|---|---|---|
| R3M | 5.3 | 34.5 | 0.3 | 0.0 | 0.5 | 1.2 | 0.3 | 0.0 |
| 2D BC | 10.6 | 31.2 | 3.6 | 0.4 | 4.4 | 22.8 | 8.0 | 3.6 |
| 2D BC-RNN | 10.0 | 30.0 | 3.2 | 0.0 | 0.8 | 24.0 | 4.0 | 8.0 |
| SGRv2 (ours) | **26.0** | 81.2 | 37.9 | 2.8 | 6.7 | 27.9 | 16.1 | 9.7 |

Table 8: Performance of additional baselines on MimicGen with 50 demonstrations.

recognize that SGRv2 and methods like BC-RNN, which model temporal dependencies, are complementary. Integrating temporal dependencies into SGRv2 presents a promising avenue for future research.

# E  Robustness to Visual Distractors

As illustrated in Section 4.2, the predicted per-point weights of SGRv2 effectively focus on locations that align well with object affordances. This raises the question: does this emergent capability make the model robust to distractors in the scene that it has never seen before? In Section 4.3, we present experiments involving distractor cups of different colors; nonetheless, it remains to be seen how SGRv2 performs in the presence of completely unseen objects and whether it will disregard them.

To address this question, we conduct additional experiments where we randomly introduce task-irrelevant objects (such as YCB [63] objects, basketballs, etc.) as visual distractors into the RL-Bench environments for 3 tasks (`meat off grill`, `phone on base`, and `push buttons`). Refer to Figure 7 for a visualization. We then test the SGRv2 and RVT models, which were previously trained on data without distractors, in these modified environments. The results, as shown in Table 10, indicate that SGRv2 is more robust to these distractors compared to RVT. This suggests that SGRv2 can effectively focus on relevant areas even in the presence of unseen objects, demonstrating robustness to visual distractors.

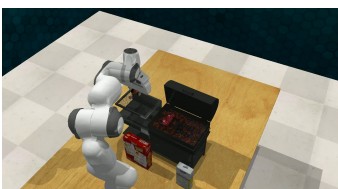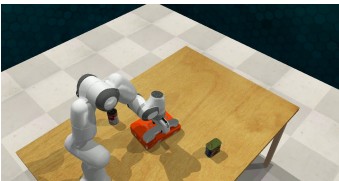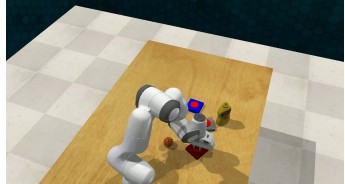

Figure 7: RLBench tasks with visual distractors.

# F  Detailed Results

For our simulation experiments using the SGRv2 on RLBench with 5 demonstrations (mentioned in Table 1) and on ManiSkill2 and MimicGen with 50 demonstrations (mentioned in Table 2), we employed 3 random seeds to ensure the reliability of our results. In the main body of the paper, we present averaged results for clarity. Here we include both the mean and standard deviation derived from our simulation results. The results for RLBench are shown in Table 11, and the results for ManiSkill2 and MimicGen are presented in Table 12.

We also report the ablations mentioned in Table 3 for each task in Table 13.

| Method | Avg. Success ↑ | Stack | StackThree | Square | Threading | Coffee | HammerCleanup | MugCleanup |
|---|---|---|---|---|---|---|---|---|
| 2D BC | 21.9 | 62.8 | 23.6 | 14.8 | 14.4 | 4.8 | 24.8 | 8.4 |
| 2D BC-RNN | 41.1 | 84.0 | 51.6 | 15.2 | 16.8 | 69.6 | 22.4 | 28.4 |
| SGRv2 (ours) | **55.8** | 95.2 | 80.4 | 32.4 | 42.2 | 74.4 | 38.0 | 28.2 |

Table 9: Performance on MimicGen with 200 demonstrations.

| | Meat Off Grill | Phone On Base | Push Buttons |
|---|---|---|---|
| RVT on env w/o distractors | 90.5 | 62.3 | 90.4 |
| RVT on env w/ distractors | 65.0 | 2.5 | 67.5 |
| SGRv2 on env w/o distractors | 96.5 | 84.1 | 93.2 |
| SGRv2 on env w/ distractors | 92.4 | 80.4 | 81.7 |

Table 10: Performance evaluation in environments with and without visual distractors.

| Method | Avg. Success ↑ | Open Microwave | Open Door | Water Plants | Toilet Seat Up | Phone On Base | Put Books | Take Out Umbrella | Open Fridge |
|---|---|---|---|---|---|---|---|---|---|
| R3M | 4.7 | 0.9 ± 0.6 | 36.4 ± 3.7 | 2.9 ± 3.7 | 15.5 ± 2.1 | 0.0 ± 0.0 | 0.5 ± 0.9 | 5.2 ± 8.7 | 3.2 ± 1.4 |
| PointNeXt | 25.3 | 7.1 ± 6.3 | 60.9 ± 5.2 | 5.6 ± 4.6 | 49.9 ± 14.7 | 46.4 ± 4.9 | 57.5 ± 8.2 | 37.5 ± 2.3 | 9.2 ± 4.0 |
| PerAct | 22.3 | 4.3 ± 7.0 | 59.6 ± 16.0 | 28.5 ± 3.1 | 69.3 ± 11.1 | 0.0 ± 0.0 | 25.1 ± 4.4 | 75.9 ± 7.0 | 3.1 ± 1.3 |
| SGR | 23.6 | 6.4 ± 2.2 | 55.3 ± 3.7 | 24.9 ± 8.2 | 30.7 ± 9.2 | 47.2 ± 1.4 | 29.3 ± 5.2 | 36.3 ± 6.4 | 7.1 ± 1.5 |
| RVT | 40.4 | 18.3 ± 1.8 | 71.2 ± 2.8 | 34.8 ± 3.3 | 47.6 ± 6.7 | 62.3 ± 1.4 | 46.5 ± 10.9 | 85.3 ± 4.5 | 24.0 ± 4.2 |
| SGRv2 (ours) | **53.2** | 27.2 ± 2.0 | 76.8 ± 8.0 | 38.0 ± 1.7 | 89.6 ± 2.8 | 84.1 ± 4.5 | 63.7 ± 11.8 | 74.5 ± 5.5 | 13.2 ± 3.4 |

| Method | Open Drawer | Slide Block | Sweep To Dustpan | Meat Off Grill | Turn Tap | Put In Drawer | Close Jar | Drag Stick | Stack Blocks |
|---|---|---|---|---|---|---|---|---|---|
| R3M | 0.0 ± 0.0 | 24.0 ± 8.8 | 0.4 ± 0.4 | 0.1 ± 0.2 | 26.1 ± 7.2 | 0.0 ± 0.0 | 0.0 ± 0.0 | 0.3 ± 0.5 | 0.0 ± 0.0 |
| PointNeXt | 21.7 ± 20.4 | 59.5 ± 22.1 | 42.0 ± 34.7 | 59.9 ± 17.8 | 48.7 ± 13.4 | 17.1 ± 27.8 | 36.0 ± 4.6 | 18.5 ± 32.1 | 1.9 ± 1.6 |
| PerAct | 56.4 ± 18.0 | 47.5 ± 24.3 | 2.8 ± 0.4 | 85.9 ± 6.9 | 8.0 ± 7.5 | 0.1 ± 0.2 | 0.5 ± 0.6 | 10.3 ± 6.4 | 1.7 ± 0.6 |
| SGR | 31.9 ± 6.2 | 72.0 ± 27.1 | 43.6 ± 8.4 | 52.7 ± 5.1 | 34.4 ± 7.4 | 8.3 ± 9.2 | 13.3 ± 5.6 | 64.4 ± 11.4 | 0.0 ± 0.0 |
| RVT | 75.1 ± 2.6 | 85.1 ± 2.2 | 19.6 ± 17.4 | 90.5 ± 2.2 | 38.4 ± 5.4 | 19.6 ± 5.5 | 25.2 ± 3.6 | 45.7 ± 10.9 | 8.8 ± 4.2 |
| SGRv2 (ours) | 81.3 ± 3.1 | 100.0 ± 0.0 | 61.5 ± 7.2 | 96.5 ± 3.9 | 87.9 ± 6.9 | 75.9 ± 3.6 | 25.6 ± 2.2 | 94.9 ± 0.6 | 17.5 ± 3.0 |

| Method | Screw Bulb | Put In Safe | Place Wine | Put In Cupboard | Sort Shape | Push Buttons | Insert Peg | Stack Cups | Place Cups |
|---|---|---|---|---|---|---|---|---|---|
| R3M | 0.0 ± 0.0 | 0.3 ± 0.2 | 0.4 ± 0.4 | 0.0 ± 0.0 | 0.0 ± 0.0 | 6.8 ± 3.7 | 0.0 ± 0.0 | 0.0 ± 0.0 | 0.0 ± 0.0 |
| PointNeXt | 4.1 ± 1.5 | 12.1 ± 4.2 | 31.5 ± 4.5 | 3.3 ± 0.6 | 0.4 ± 0.4 | 22.0 ± 38.1 | 0.1 ± 0.2 | 4.4 ± 3.8 | 0.4 ± 0.4 |
| PerAct | 4.4 ± 5.2 | 0.9 ± 0.9 | 8.7 ± 1.7 | 0.4 ± 0.4 | 0.4 ± 0.4 | 83.1 ± 5.3 | 1.9 ± 1.2 | 0.1 ± 0.2 | 0.7 ± 0.6 |
| SGR | 0.9 ± 0.8 | 16.9 ± 2.2 | 24.7 ± 5.8 | 0.1 ± 0.2 | 0.1 ± 0.2 | 12.0 ± 1.4 | 0.1 ± 0.2 | 0.0 ± 0.0 | 1.1 ± 0.9 |
| RVT | 24.0 ± 3.8 | 30.7 ± 4.9 | 92.7 ± 0.6 | 5.6 ± 2.1 | 1.6 ± 0.7 | 90.4 ± 2.9 | 4.0 ± 0.0 | 3.1 ± 0.6 | 1.2 ± 0.7 |
| SGRv2 (ours) | 24.1 ± 0.6 | 55.6 ± 8.0 | 53.1 ± 7.4 | 20.3 ± 9.2 | 1.9 ± 0.6 | 93.2 ± 5.3 | 4.1 ± 1.4 | 21.3 ± 11.8 | 1.6 ± 0.7 |

Table 11: RLBench results (%) on 5 demonstrations with mean and standard deviation.

| Method | Avg. Success ↑ | Avg. Rank ↓ | LiftCube | PickCube | StackCube | PickSingleYCB |
|---|---|---|---|---|---|---|
| PointNeXt | 16.8 | 2.5 | 50.8 ± 15.2 | 4.7 ± 0.4 | 10.6 ± 4.3 | 1.1 ± 0.1 |
| SGR | 14.9 | 2.5 | 26.9 ± 4.0 | 12.2 ± 3.1 | 3.5 ± 2.2 | 17.0 ± 0.2 |
| SGRv2 (ours) | **55.8** | **1.0** | 80.5 ± 7.3 | 72.9 ± 4.1 | 27.7 ± 4.3 | 42.2 ± 2.3 |

| Method | Avg. Success ↑ | Avg. Rank ↓ | Stack | StackThree | Square | Threading | Coffee | HammerCleanup | MugCleanup |
|---|---|---|---|---|---|---|---|---|---|
| PointNeXt | 13.6 | 2.9 | 56.1 ± 6.4 | 3.7 ± 1.4 | 0.9 ± 0.5 | 3.6 ± 2.2 | 12.0 ± 5.2 | 11.7 ± 2.8 | 7.1 ± 0.9 |
| SGR | 14.2 | 2.0 | 50.8 ± 7.7 | 5.6 ± 1.7 | 1.3 ± 0.5 | 4.0 ± 0.8 | 14.1 ± 2.0 | 14.1 ± 1.7 | 9.7 ± 2.4 |
| SGRv2 (ours) | **26.0** | **1.0** | 81.2 ± 4.4 | 37.9 ± 1.5 | 2.8 ± 0.7 | 6.7 ± 2.0 | 27.9 ± 7.0 | 16.1 ± 3.9 | 9.7 ± 2.7 |

Table 12: ManiSkill2 and MimicGen results (%) on 50 demonstrations with mean and standard deviation.

| Method | Avg. Success ↑ | Open Microwave | Open Door | Water Plants | Toilet Seat Up | Phone On Base | Put Books | Take Out Umbrella | Open Fridge |
|---|---|---|---|---|---|---|---|---|---|
| SGRv2 | 53.2 | 27.2 | 76.8 | 38.0 | 89.6 | 84.1 | 63.7 | 74.5 | 13.2 |
| SGRv2 w/o decoder | 21.3 | 4.4 | 68.4 | 12.4 | 38.8 | 32.8 | 27.2 | 35.6 | 6.8 |
| SGRv2 w/ absolute pos prediction | 21.0 | 8.0 | 57.6 | 21.2 | 17.2 | 14.8 | 21.2 | 44.4 | 4.4 |
| SGRv2 w/ uniform point weight | 44.2 | 21.6 | 82.4 | 28.8 | 32.0 | 60.4 | 68.0 | 44.0 | 14.0 |
| SGRv2 w/o dense supervision | 40.1 | 6.4 | 59.2 | 6.8 | 54.4 | 78.4 | 60.8 | 68.0 | 6.0 |

| Method | Open Drawer | Slide Block | Sweep To Dustpan | Meat Off Grill | Turn Tap | Put In Drawer | Close Jar | Drag Stick | Stack Blocks |
|---|---|---|---|---|---|---|---|---|---|
| SGRv2 | 81.3 | 100.0 | 61.5 | 96.5 | 87.9 | 75.9 | 25.6 | 94.9 | 17.5 |
| SGRv2 w/o decoder | 14.0 | 78.8 | 15.6 | 40.8 | 46.8 | 4.0 | 18.8 | 0.8 | 0.0 |
| SGRv2 w/ absolute pos prediction | 20.4 | 99.2 | 50.8 | 10.8 | 68.8 | 4.0 | 6.4 | 54.8 | 1.2 |
| SGRv2 w/ uniform point weight | 92.8 | 100.0 | 72.8 | 90.8 | 67.6 | 74.0 | 43.6 | 97.6 | 1.6 |
| SGRv2 w/o dense supervision | 75.2 | 92.8 | 19.2 | 72.0 | 84.0 | 42.0 | 28.8 | 59.6 | 43.2 |

| Method | Screw Bulb | Put In Safe | Place Wine | Put In Cupboard | Sort Shape | Push Buttons | Insert Peg | Stack Cups | Place Cups |
|---|---|---|---|---|---|---|---|---|---|
| SGRv2 | 24.1 | 55.6 | 53.1 | 20.3 | 1.9 | 93.2 | 4.1 | 21.3 | 1.6 |
| SGRv2 w/o decoder | 3.2 | 16.4 | 27.6 | 0.0 | 0.0 | 56.8 | 0.4 | 0.8 | 1.6 |
| SGRv2 w/ absolute pos prediction | 0.4 | 1.6 | 3.2 | 0.0 | 0.0 | 35.2 | 1.6 | 0.0 | 0.0 |
| SGRv2 w/ uniform point weight | 1.6 | 32.8 | 50.4 | 1.2 | 0.8 | 64.4 | 0.0 | 5.2 | 0.8 |
| SGRv2 w/o dense supervision | 0.0 | 30.4 | 52.0 | 0.4 | 0.0 | 100.0 | 0.0 | 0.0 | 2.4 |

Table 13: Ablations results (%) for SGRv2 on RLBench with metrics for each task.

