# OpenReview forum: "Leveraging Locality to Boost Sample Efficiency in Robotic Manipulation"
_robot-learning.org/CoRL/2024/Conference — CoRL 2024_

### Official Review · Reviewer_tgDw · 2024-07-17
**Straight-forward method exploiting inductive bias in robotics to improve performance and lower data requirement.**

**Originality:** 4
**Technical Quality:** 4
**Clarity Of Presentation:** 5
**Potential Impact:** 3
**Recommendation:** 3
**Confidence:** 4

**Review:**

Pros
- Very well-written manuscript, easy to follow, nice organization.
- Using pixel-wise dense prediction has been shown effective in many 3D vision tasks so it’s worth trying that in robotics as well. The emergent property on object affordances looks interesting.
- Experimental result looks strong comparing to the baselines.

Cons
- I suggest giving a clearer definition early in the paper, of what locality and sample efficiency the authors are looking for. E.g. training efficiency/fewer data, locality as in action representation/object affordances.
- Figure 3 is bad visually, could be improved. Is that a chair? It would be helpful to list the task there too.
- In Sec 4.3, SGRv2 w/o sem - if the semantic branch is removed, the input to the network is only the point cloud? If so how does it recognize the color for seen/unseen scenarios, or is it just overfit to the trained action sequence. And if the color image is part of the input, there is no reason that it will generalize to unseen color tasks.

**Quality Of The Limitations Section:**

3

**Questions For Rebuttal:**

- Does the proposed method improve generalization to unseen object configurations, such as different object position/orientation/view point? For example in RLBench the train/test variance is very small. I imagine the relative encoding and use of 3D/point cloud would help a bit on that.
- For the real setup, long-horizon tasks, are the testing setups identical to the training ones?
- Are the models trained per-task or is there one model for all the tasks, with 5 demonstrations each?

**Robotics Focus:**

4

**Summary Of Paper:**

SGRv2 is an imitation learning framework that try to leverage the locality in robot actions to improve sample efficiency. The framework uses dense point-wise feature and predicts actions using the weighted average from such dense features. Another design choice is to encode the action in a relative coordination frame. With these inductive biases SGRv2 achieves higher task success rate using less training data.

**Summary Of Recommendation:**

Overall I think it’s a nice paper and recommend accepting. The method used here is quite straight-forward and demonstrates strong empirical result.

---

### Official Review · Reviewer_AWsF · 2024-07-20

**Originality:** 3
**Technical Quality:** 4
**Clarity Of Presentation:** 4
**Potential Impact:** 3
**Recommendation:** 3
**Confidence:** 4

**Review:**

Strengths:

* This work tackles the important problem of improving the sample efficiency of point cloud based imitation learning methods.
* The paper is well-written and easy to follow. The figure assists gaining good understanding of the paper.
* Although the idea of leveraging locality via aggregating per-point predictions is not new, the paper shows a novel integration of such idea with prior work and good performance with comprehensive experiments.
* Simulation results are convincing. Ablation results are thoughtfully designed and nicely executed.
* Real robot experiments show that proposed method can achieve good performance even with less than 10 demonstrations.

Weakness and questions:

* What is the training and inference speed of the method?
* What do the $w_*(f_i)$ MLPs take as input? Is there any direct supervision on these weights? How are the per-point predictions aggregated for rotation, gripper open state, and collision indicator?
* Since RVT is the second best performing method in Table 1, it would be nice to include this baseline in ManiSkill2 and MimicGen experiements too unless there is a reason that prevents such a comparison. It would be also nice to add a curve for RVT in the left plot of Figure 1 if possible.
* If the predicted per-point weights of the model can focus on places with that align well with object affordances, is the method robust to distractors in the scene that it has never scene before? In the "Generalization Task" section, the authors showed experiments with distractor cups, but what if there are objects that are completely unseen? Will the method ignore them?
* On the website of the paper, it would be good to add visualizations of demonstrations as well as the distribution of initial poses for objects in these demonstrations. This will facilitate better understanding of the real robot setup.
* It would be great to discuss about failure cases of the proposed method in real robot experiments.

Minor correction:

* Table 2 caption: "buttom" -> "bottom"

**Quality Of The Limitations Section:**

3

**Questions For Rebuttal:**

See above.

**Robotics Focus:**

4

**Summary Of Paper:**

This paper presents a visuomotor policy that incorporates novel visual and action representations for sample efficient imitation learning. The proposed method, SGRv2, builds upon its predecessor SGR but introduces the use of local per-point features. Instead of using global features to predict actions, SGRv2 uses an encoder-decoder architecture to extract per-point features from point cloud inputs. The method then predicts actions relative the end-effector at each point and aggregates these per-point predictions with a learned a weighted average.

**Summary Of Recommendation:**

This work presents an interesting method to improve sample efficiency of visuomotor policies for imitation learning. The method is not ground-breaking but are well-designed and accompanied with convincing results. Some additional comparisons and analysis can make this paper stronger.

---

### Official Review · Reviewer_LbA4 · 2024-07-23

**Originality:** 2
**Technical Quality:** 2
**Clarity Of Presentation:** 3
**Potential Impact:** 3
**Recommendation:** 2
**Confidence:** 4

**Review:**

**Strengths**

- The paper is well-written and the method description and experiments are easy to follow.
- The proposed changes show a clear advantage over the baseline SGR method
- Real robot experiments are provided.

**Weaknesses**

- The experiment scope is limited and does not paint a complete picture of the performance of the proposed method compared to other baselines. The experiment setup seems carefully designed to make the proposed method outperform other baselines. For example, in RLBench, all methods are evaluated on 5 demonstrations, and in MimicGen and ManiSkill2, methods are evaluated on 50 demos. It would make more sense to evaluate all methods on more settings (e.g. different numbers of demonstrations) to present a more complete understanding of the advantages of the proposed approach.

- In some cases, comparison against more baselines would be highly valuable. For example, the MimicGen paper uses visuomotor policies without depth, and has a complete set of results for using 1000 demonstrations on all tasks. Comparing all methods in that setting would be useful. Comparing similar visuomotor policies against the proposed method on 50 demonstrations would also be useful, to contextualize the performance in this setting. Some of the MimicGen numbers reported in this paper appear low. For example, Threading has 6.7 success on 50 demos, but in the MimicGen paper, Threading has 19.3 on the 10 source human demos using a visuomotor policy. Similarly, Coffee has 27.9 success rate on 50 demos, but in the MimicGen paper, Coffee has 74 on the 10 source human demos. There are other suitable baselines for closed-loop control as well in this setting, such as Diffusion Policy (https://diffusion-policy.cs.columbia.edu/) and 3D Diffusion Policy (https://arxiv.org/abs/2403.03954) (for a comparison against a closed-loop policy that uses depth information).

- The overall performances reported in the paper are low. This is reasonable given the limited number of demonstrations, but this makes the significance of the presented experiment settings limited, especially since the evaluated benchmarks provide ways to procedurally generate more demonstrations for each task.

**Additional Comments**

- Typo in Table 2 caption (says "buttom")
- In Figure 2, should add more details on method, e.g. what the semantic branch, geometric branch, etc. entails

**Quality Of The Limitations Section:**

3

**Questions For Rebuttal:**

See the points raised above.

**Robotics Focus:**

4

**Summary Of Paper:**

This paper proposes SGRv2, an imitation learning method that is sample efficient in the number of demonstrations used. The method builds on prior work (SGR), with several improvements added. The method is shown to improve over other baselines when trained on limited quantities of demonstration data, and is also demonstrated on real hardware.

**Summary Of Recommendation:**

More thorough evaluations are needed to assess the quality of the proposed method.

---

### Author Rebuttal · Authors · 2024-08-10

As suggested by Reviewer LbA4, we tested SGRv2 and the baseline under conditions with visual distractors. Here we upload three videos showing the modified environments where visual distractors were applied.

---

### Decision · Program_Chairs · 2024-09-04

**Decision:**

Accept

**Comment:**

The reviewers found some strengths in this submission, but also clearly articulated some questions for the rebuttal phase.
Thank you for your detailed responses.